# Calcium Signalling in Heart and Vessels: Role of Calmodulin and Downstream Calmodulin-Dependent Protein Kinases

**DOI:** 10.3390/ijms232416139

**Published:** 2022-12-17

**Authors:** Sofia Beghi, Malgorzata Furmanik, Armand Jaminon, Rogier Veltrop, Nikolas Rapp, Kanin Wichapong, Elham Bidar, Annamaria Buschini, Leon J. Schurgers

**Affiliations:** 1Cardiovascular Research Institute Maastricht (CARIM), Department of Biochemistry, Maastricht University, P.O. Box 616, 6200 MD Maastricht, The Netherlands; 2Department of Chemistry, Life Sciences and Environmental Sustainability, University of Parma, Parco Area Delle Scienze 11A, 43124 Parma, Italy; 3Department of Cardiothoracic Surgery, Heart and Vascular Centre, Maastricht University Medical Centre+, 6229 HX Maastricht, The Netherlands

**Keywords:** cardiovascular disease, calcium signalling, calmodulin, calcium calmodulin dependent protein kinases

## Abstract

Cardiovascular disease is the major cause of death worldwide. The success of medication and other preventive measures introduced in the last century have not yet halted the epidemic of cardiovascular disease. Although the molecular mechanisms of the pathophysiology of the heart and vessels have been extensively studied, the burden of ischemic cardiovascular conditions has risen to become a top cause of morbidity and mortality. Calcium has important functions in the cardiovascular system. Calcium is involved in the mechanism of excitation–contraction coupling that regulates numerous events, ranging from the production of action potentials to the contraction of cardiomyocytes and vascular smooth muscle cells. Both in the heart and vessels, the rise of intracellular calcium is sensed by calmodulin, a protein that regulates and activates downstream kinases involved in regulating calcium signalling. Among them is the calcium calmodulin kinase family, which is involved in the regulation of cardiac functions. In this review, we present the current literature regarding the role of calcium/calmodulin pathways in the heart and vessels with the aim to summarize our mechanistic understanding of this process and to open novel avenues for research.

## 1. Introduction

Cardiovascular disease (CVD) is a complex disease with many factors and events involved. The disbalanced regulation of physiology, mechanobiology, and molecular biology results in heart and vessel disease. Indeed, different factors coexist and cooperate, together increasing the risk of initiation and progression of CVD [1,2,3]. CVD is the leading cause of death all over the world, and more people die annually from this type of disease than from any other cause. According to the World Health Organization, every year 17 million people die of CVD globally. Additionally, in 2020, in which the death rate increased due to COVID-19, CVD remained the principal cause of death [4]. Among CVD, atherosclerosis is the most common cause, which is associated with the growth and build-up of cholesterol-rich plaques within the inner wall of the vessel. Arterial occlusion can limit or block blood flow to the heart or other organs, causing myocardial infarction, ischaemic cardiomyopathy, stroke and peripheral arterial disease [5].

Among the different pathways involved in cardiovascular homeostasis, the regulation of calcium signalling plays a crucial role. One of the major pathways in which calcium is involved is excitation contraction coupling. Here, the rise of intracellular calcium (Ca^2+^) in cardiomyocytes and vascular smooth muscle cells (VSMCs) is sensed by calmodulin (CaM), a multifunctional intermediate target of the secondary messenger Ca^2+^. CaM ensures activation of downstream calcium calmodulin kinase proteins (CaMKs), including CaMKK1, CAMKK2, CaMKI, CaMKII, and CaMKIV. CaMKs are kinases that upon phosphorylation result in multiple cellular functions. This signalling pathway is called the Ca^2+^-CaM-dependent kinase cascade [6,7] and is involved in many cellular processes, including glucose homeostasis, apoptosis, hematopoietic stem cell maintenance, normal immune cell function and cell proliferation [8,9,10,11,12].

Atherosclerosis as well as aortic aneurysm are aggravated by dysregulated calcium metabolism, causing vascular calcification (VC). VC is a pathological condition caused by calcium-phosphate precipitation leading to vascular stiffness. VC is an actively regulated process with a key role for VSMCs, the major cell type in the media layer of arteries. VSMCs are involved in regulating vascular tone and remodelling processes in the vessel wall upon injury [13,14,15]. VC is driven by an imbalance in inhibitors (e.g., matrix gla protein) and promotors (e.g., Ca^2+^/PO_4_^3−^, cell death, VSMC dedifferentiation and extracellular matrix remodelling) [16]. Calcium deposition leading to calcification takes place extracellularly. However, its nidus may be derived from intracellular signalling leading to extracellular vesicle release from VSMCs [17]. Similarly, calcium-phosphate deposition in aortic heart valves causes aortic valve insufficiency and is mediated by vascular interstitial cells that create the nidus for calcium-phosphate crystals in the extracellular space. Aortic valve calcification narrows the opening of the valve, reducing blood flow and is linked to aortic valve stenosis [18,19,20].

Current knowledge of CVD and the underlying mechanisms is insufficient to decrease the burden of disease. The aim of our review is to summarize the state-of-the-art knowledge on the role of calcium signalling in cardiac and vascular tissue, with a focus on calmodulin and its downstream calcium kinases.

### The Role of Calcium in the Cardiovascular System

Calcium is an abundant element in nature and the most abundant mineral in the human body. Calcium constitutes 1.43% of the human body, whilst phosphate makes up 1.11% of the human body. All other minerals and metals accumulate to a combined 0.73% of the human body. Some 99% of calcium in the body is found in bones and teeth, whilst the other 1% is found in blood and soft tissues. Calcium plays a crucial role in the heart and vessels, both in physiological and pathological conditions [7,21,22,23,24].

Cells in the heart and blood vessels need calcium to contract and to perform their function. The concentration of calcium within the cytosol is some 100 nM. The calcium concentration outside the cell is some 2 mM, thus a factor of 20,000 higher. Therefore, the influx of calcium into the cell needs to be tightly regulated. This calcium entry is regulated by calcium-sensing receptors (CaSR), which are essential to sense Ca^2+^ in the extracellular fluid and facilitate active and orchestrated calcium uptake [25,26,27]. Upon calcium entry, this activates a series of downstream pathways that directly regulate cellular functions [21,24]. The concentration of calcium inside cardiac cells has a heterogeneous distribution thanks to the presence of different microdomains in the cytosol, the spaces between intracellular organelles, and the delimited compartments such as the sarcoplasmic reticulum. Calcium homeostasis and signalling within the cell are regulated by calcium channels that come in many forms and flavours with diverse functions and structures. The main role of calcium channels is to balance both the cellular influx and the efflux of calcium [28,29]. Calcium channels can be stimulated by numerous and diverse stimuli such as membrane depolarization, extracellular and intracellular signalling molecules, and physical forces such as stretching or temperature. Among the calcium channels on plasma membranes, the voltage-gated calcium channels are one of the most important as they rapidly transport calcium into the cytoplasm. These channels respond to a change in voltage across the cell membrane. The plasma membrane also harbours calcium transport mechanisms involved in removing calcium from the cell to provide a low cytosolic concentration. These calcium transport mechanisms include, amongst others, the plasma membrane calcium ATPase (PMCA) pump and the Na^+^/Ca^2+^ exchanger (NCX) [24,30,31].

Calcium is an essential central player in the heart because it is involved in the core function of excitation–contraction coupling (ECC) [23]. ECC is a number of events starting at the production of the action potential (electrical impulse) to the contraction of the heart muscles. Additionally, calcium is a co-factor for many enzymes, including proteins involved in blood coagulation. Platelets and several coagulation factors are activated by Ca^2+^, which is responsible for maintaining blood clot architecture and strength [32,33]. Calcium is also involved in the regulation of long-term processes, such as gene regulation, in order to change protein expression. Moreover, calcium has a central role in adaptive tissue regulation, such as cardiac hypertrophy [22].

In VSMC physiology, calcium fulfils a pivotal role in physiology as well as in pathology. Here, calcium is a key element in regulating blood flow and vascular luminal diameter. Calcium induces contraction of VSMCs by creating a complex with the ubiquitous calcium binding protein calmodulin and increasing the activity of myosin light-chain kinase [34]. VSMCs are characterized by a high degree of plasticity, having the capability to switch phenotype. Indeed, under physiological conditions, VSMCs are in a quiescent contractile phenotype. However, upon pathobiological or mechanical stress, VSMCs switch towards a synthetic phenotype, characterized by a higher degree of migration and proliferation [14,35,36]. In both VSMC phenotypes, intracellular calcium plays a different but pivotal role. The influx of calcium in VSMC with contractile phenotype is regulated mainly via voltage-dependent L-type calcium channels, responsible for excitation–contraction coupling. Upon mechanical or biological stress, VSMCs switch phenotype to a reparative and synthetic VSMC. Synthetic VSMCs shed extracellular vesicles that contain endogenous calcification inhibitors to prevent mineralization of the extracellular matrix. Upon long-term stress and mineral disbalance, the content of these extracellular vesicles changes, turning them from anti-calcific into pro-calcific vesicles [37]. These extracellular vesicles form the nidus for VC and their uptake by synthetic VSMCs causes a rise in intracellular calcium and subsequently cellular stress, eventually leading to cell death. In synthetic VSMCs, calcium entry is less dependent on voltage-dependent L-type calcium channels [34]. Instead, calcium entry is in part via clathrin-mediated extracellular vesicles and may contribute to vascular calcification [36]. Intracellular calcium levels in VSMCs are very dynamic, especially in contractile VSMC, where calcium changes both in spatial and temporal domains. These oscillations of intracellular calcium are more pronounced in contractile VSMCs compared to VSMCs with a synthetic phenotype. The higher oscillation frequencies in contractile VSMC can increase kinase activity in a frequency-dependent manner through a trapping phenomenon consisting of the acquisition of autonomous kinase activity of certain CaM kinases [34].

Below we describe the role of calmodulin and its signalling pathways in more detail in relation to CVD.

## 2. Calmodulin

Calmodulin (CaM) is involved in the regulation and transduction of calcium signalling. CaM is an important primary sensor of intracellular calcium levels in eukaryotic cells, playing a pivotal role in the transduction and deciphering of calcium signalling [38,39]. CaM is a low-molecular-weight protein of 16 kDa, composed of 149 amino acid residues. There are three different human genes (*CaM1*, *CaM2*, *CaM3*) that encode for three highly conserved proteins (differing only at the nucleotide level) [40,41,42,43]. Each gene is characterized by distinct promoter elements and by unique 5′- and 3′-untranslated regions that may permit regulation of CaM expression at discrete cellular sites during differentiation [44]. CaM is an α-helical protein composed of N- and C-terminal lobes, each containing two calcium-binding EF-hands. Thus, a total of four Ca^2+^ can be bound per CaM (Figure 1) [38,39,40]. Calcium binding to CaM induces a conformational change, which is unequivocal for activation and modulation of second messenger downstream proteins such as adenylate cyclase, serine/threonine kinases, nitric oxide synthase, serine threonine protein phosphatases (i.e., calcineurin), which are all involved in the transduction of intracellular calcium signalling (Figure 2). CaM also regulates calcium transport via plasma membrane ATPase, a high-affinity calcium pump that contributes to the maintenance of intracellular calcium homeostasis [45]. CaM also binds to several structural proteins such as spectrin, neuromodulin, and caldesmon [43,44]. These proteins have been shown to play important roles in the maintenance of plasma membrane integrity [46] and cytoskeletal structure [47], postsynaptic function, and in the regulation of smooth muscle cell contraction, respectively [48]. As a primary sensor of intracellular calcium levels, CaM plays a key role in a multitude of processes, such as the regulation of cell cycle, fertilization, intracellular signalling, differentiation, cell death, and cell contraction [39,49].

The main function of CaM in the heart is modulation of the action potential, leading to rapid contraction of a distinct group of cardiac cells through the regulation of different channels [22,38], such as the voltage-gated sodium channels [50], the voltage gated potassium channels [51], and the voltage-gated calcium channels [52]. CaM is also involved in the regulation of the sarcoplasmic reticulum calcium release channel (RyR2), the main source of intracellular calcium necessary to induce contraction [53,54]. Regulation of RyRs by CaM is isoform specific. For RyR1, CaM exhibits biphasic regulation that depends on Ca^2+^ concentration. It acts as a weak activator at nanomolar concentrations of Ca^2+^ (apo-CaM) and as an inhibitor at micromolar concentrations of Ca^2+^ (Ca^2+^-CaM). On the other hand, for RyR2, Ca^2+^-CaM only inhibits the channel, without activating effects [55]. Moreover, CaM interacts with secondary pathway effectors of cardiomyocyte contraction, such as the beta-adrenergic pathway [56,57] and cyclic nucleotide signalling [38]. In the context of the modulation of cyclic nucleotide signalling by CaM, it is important to consider the role of the phosphodiesterases (PDEs), which are functional antagonists of the adenyly cyclases (ACs) as they degrade cyclic nucleotides and subsequently lower, for example, the cAMP-dependent protein kinase-A (PKA) activity. Among the 21 mammalian genes of PDE, 3 PDE1 gene products (PDE1A–C) appear particularly interesting as PDE1A–C are all CaM regulated and found in heart tissue. The binding of Ca^2+^-CaM to N-terminal sites in PDE1 stimulates their activity and thus increases cell cAMP and cGMP degradation [58]. Degradation of cyclic nucleotides generally signals an easing of cardiomyocyte contractility. Finally, CaM interacts with downstream calcium CaM dependent kinases, supporting calcium recycling in cardiomyocytes necessary to maintain calcium homeostasis in preparation for a new excitation event [38].

During evolution, the number of functions for CaM has significantly increased with the need for sequence conservation [59]. In fact, CaM is highly conserved, and it has been shown that most of the human calmodulin mutations are deleterious and associated with life-threatening conditions during early infancy. Defects in CaM function disrupt important calcium signalling events, possibly resulting in fatal heart disease [44,45,59,60,61] (Table 1). Indeed, deleterious variants in CaM genes (*CaM1*, *CaM2*, and *CaM3*) cause calmodulinopathy, a rare life-threatening arrhythmia syndrome [59,61]. The term specifically refers to a wide spectrum of clinical manifestations [61], such as long-QT syndrome (LQTS) [59,62,63,64,65], catecholaminergic polymorphic ventricular tachycardia (CPVT) [66], or idiopathic ventricular fibrillation (IVF) [67]. CaM mutations were also identified in autopsy-negative sudden unexplained deaths (SUD) in young individuals [61,68]. Three heterozygous de novo mutations in either *CaM1* or *CaM2* were identified that caused an alteration of residues near the calcium binding loops in the calmodulin carboxyl-terminal domain. These alterations were observed in infants who exhibited life-threatening ventricular arrhythmias combined variably with epilepsy and delayed neurodevelopment [59]. Moreover, a common mutation in a highly conserved residue (Phe90) in CaM1 resulted in underlying IVF manifestation in childhood and adolescence [67]. Further, a novel missense mutation in CaM1 was identified in a Moroccan family with a history of ventricular tachycardia and sudden death. This missense mutation was associated with exercise-induced QT prolongation, ventricular tachycardia, and sudden cardiac death during childhood [69].

In VSMCs, CaM is a critical calcium sensor which regulates different downstream proteins. The Ca^2+^-CaM complex is necessary to phosphorylate and activate myosin light-chain (MLC) actin–myosin interactions and VSMC contractions [70,71,72]. Specifically, CaMKII phosphorylates and inactivates MLC kinase, a process that may be essential in the regulation of VSM contraction [70]. CaM-calcium binding also controls the interaction between alpha-smooth muscle actin (αSMA) and myosin, affecting VSMC elasticity and adhesion processes [73]. Further, Ca^2+^-CaM activates the serine/threonine (Ser/Thr) phosphatase calcineurin as well as the family of Ca^2+^-CaM-dependent protein kinases, all involved in the regulation of cell cycle progression [70]. The role of CaM in VSMC-driven vascular disease (i.e., atherosclerosis, aneurysms, calcification) has not been studied in detail thus far.

Taken together, these findings highlight the role and function of CaM in the cardiovascular system.

## 3. Calcium Calmodulin Kinases

The transduction and amplification of intracellular calcium signals sensed by CaM results in regulation of downstream effectors involved in phosphorylation. Specifically, proteins involved in this pathway are of the calcium calmodulin kinase protein family (CaMK), a category of enzymes all classified as Ser/Thr kinases (Figure 3). CaMKs catalyse the transfer of phosphate from the gamma position of ATP to the hydroxyl group of Ser or Thr residues in proteins [39,49,74]. CaMKs can be divided in two different groups. The first group consists of CaMKK1, CAMKK2, CaMKI, CaMKII, and CaMKIV, which are multifunctional proteins. The second group consists of CaMKIII, phosphorylase kinase, and myosin light-chain kinase and are substrate-specific kinases with only one downstream target [74]. All CaMK proteins share a common structure with a bi-lobed catalytic domain followed by a regulatory domain containing both an autoinhibitory as well as a CaM-binding site (Figure 3) [74,75]. In the case of basal intracellular calcium levels, CaMKs are in an inactivated state due to the autoinhibitory domain that interacts and blocks the CaM binding or catalytic site. When intracellular calcium levels increase, CaM binds four Ca^2+^ to become saturated, creating a conformational change. This conformational change results in interaction with the CaM binding domain in CaMKs, resulting in activation. The activation of some CaMKs (i.e., CaMKI) requires both binding of calcium as well as an additional modification, such as phosphorylation. Moreover, the activity of certain kinases (i.e., CaMKIV and CaMKII) can become independent of CaM, allowing them to be functional beyond the duration of a transient elevation in intracellular calcium [76,77,78]. These differences in regulation are important for the control of the many cellular functions of CaM.

Ca^2+^-CaM protein kinases (CaMKs) are involved in several process, controlling several different functions [75,79,80,81,82,83,84,85,86,87,88,89,90,91,92,93]. In the following sections we describe in more detail the role of multifunctional CAMKs in the heart and vasculature.

### 3.1. CAMKII

CaMKII is the most abundant calcium calmodulin kinase in the heart [7,94,95]. There are four different isoforms, namely CAMKII α, β, γ, and δ, encoded by four different genes. All four CaMKII isoforms are involved in cellular functions such as calcium homeostasis, membrane potential, the cell cycle, cytoskeletal organization, cell contraction, learning, memory, and gene expression and secretion [74,78,95]. It is important to note that the four isoforms of CaMKII have a different tissue distribution [96]. CaMKII γ and δ are mainly present in cardiac tissue whereas CaMKII α and β are expressed mainly in the neuronal system [7,57,97]. Specifically, CaMKIIδ is the predominant isoform in the heart and plays a crucial role in the regulation of several cardiac functions. CaMKIIδ is involved in the integration of calcium signalling through a variety of pathways to maintain cardiac homeostasis as it is predominantly involved in the excitation contraction coupling (ECC) in the heart [38,78,94,97,98].

The effect of CaMKII on ECC is crucial because it phosphorylates calcium handling proteins including sarcoplasmic reticulum (SR) calcium release channels, such as ryanodine receptors (RyR) [99], phospholamban (PLB) [94], and L-type calcium channels [100]. These downstream proteins of CaMKII are responsible for the regulation of cellular calcium influx, calcium release from the SR, as well as calcium uptake into the SR [7]. The phosphorylation of myofilament proteins (cMyBP-C: cardiac myosin binding protein-C) and myosin regulatory light-chain 2 by CaMKII contributes to myofilament calcium sensitization and interactions [101,102]. Tong et al. showed that the phosphorylation of cMyBP-C due to CaMKIIδ and PKA is a principal mediator of increased contractility observed with β-adrenergic stimulation or increased pacing. Moreover, its phosphorylation increases the force and kinetics of twitches in living cardiac muscle [101]. CaMKII also phosphorylates titin, which stabilizes the contractile machinery, causing a lower cardiomyocyte passive force likely associated with improved diastolic filling [103,104,105]. Meanwhile, the phosphorylation by CaMKII of troponin I enacts the calcium desensitization; this process is required to accelerate diastolic myofilament relaxation with increased heart rate [105,106]. Additionally, CaMKII targets and activates cardiac Na^+^ [107,108] and K^+^ channels [54,109,110,111], which are involved in the regulation of calcium homeostasis, through phosphorylation.

The activation of CaMKII depends on Ca^2+^-CaM binding, subsequently activating all 12 subunits in separate holoenzymes (a biochemically active compound consisting of an enzyme with its cofactor) forming the dodecameric structure of CaMKII [112]. Additionally, Ca^2+^-CaM bound to CaMKII ensures that one subunit is auto-phosphorylated at Thr286 by a neighbouring activated CaMKII subunit. This event generates autonomous activity, increasing the affinity of CaMKII for Ca^2+^-CaM some 1000-fold, an event known as “CaM-trapping” [74,77,78].

This autonomous activity of CaMKII can also be mediated by other post-translational modifications such as oxidation, nitrosylation, and glycosylation [105] The altered hyperactivate regulation of CaMKII is increased in myocardial disease, contributing to apoptosis, arrhythmias [113], defective ECC, and ETC (excitation–transcription coupling) [53,114,115], favouring pathological hypertrophy [96] and contractile dysfunction specifically during heart failure (HF) [54,96,114,115]. CaMKII is a pro-arrhythmogenic protein increased during CVD, i.e., myocardial injury [116], atrial fibrillation [117], cardiac hypertrophy, and ischaemia/reperfusion injury [118], and inhibition of CaMKII might thus be a treatment option for cardiac pathologies [7,119,120].

Additionally, in blood vessels CaMKII plays a crucial role. In VSMCs, CaMKII is activated by Ca^2+^-CaM to regulate activation of myosin light-chain kinase (MLCK) by phosphorylation [70]. In VSMCs, CaMKII inhibits CREB (cAMP response element-binding protein) through different phosphorylation sites. CREB is an important transcription factor regulating expression of contractile and proliferative and migratory genes during VSMC phenotype switching. The rise of intracellular Ca^2+^ in VSMCs promotes nuclear translocation of Ca^2+^-CaMKII. γCaMKII acts as a carrier of Ca^2+^/CaM, transporting it into the nucleus. In the nucleus Ca^2+^/CaM activates CaMKK and its substrate CaMKIV, which in turn activates CREB by phosphorylation [34,121]. Moreover, it has been shown that Ca^2+^-CaMKII and Erk1/2 activation play a crucial role in the regulation of VSMC proliferation by ⍺-adrenergic receptor agonists [122]. Finally, it was recently shown that CAMKIIδ and calponin 3 play critical roles in circRNA CDR1as/miR-7-5p-induced human pulmonary VSMC calcification under hypoxic conditions, which may contribute to pulmonary hypertension [123].

### 3.2. Ca^2+^-CaM-Dependent Kinase Cascade

#### 3.2.1. CaMKK Family

The Ca^2+^-CaM-dependent kinase cascade consists of Ca^2+^-CaM kinase-kinase family (CaMKK1 and CaMKK2), CaMKI, and CaMKIV (Figure 4). This cascade is involved in different processes, such as cell proliferation, apoptosis, immune cell function, stem cell maintenance, as well as glucose homeostasis [7]. Dysregulation of kinases in the Ca^2+^-CaM-dependent kinase cascade is associated with a variety of diseases such as cancer, obesity, diabetes, neuronal defects, and CVD [12,124].

CaMKKs support transduction of calcium to other downstream kinases such as CaMKI and CaMKIV [32,62]. They are present both in the cytoplasm as well as in the nucleus of the cell. There are two isoforms of CaMKKs encoded by the genes *CAMKK1* and *CAMKK2* [39,74,125]. Both CaMKKs share a high sequence homology and the same common domain structure typical for all kinase proteins (Figure 3) [125,126]. The main difference between both CaMKKs is that CaMKK1 is kept in an inactive state until binding of Ca^2+^-CaM relieves the autoinhibitory mechanism, whilst autophosphorylation allows a partially autonomous activity of CaMKK2 in the absence of Ca^2+^-CaM [74,75,126]. Both kinases can be partially inhibited by PKA that phosphorylates CaMKKs at different Ser or Thr residues within the CaM kinase binding domain and within the ATP binding region in the catalytic domain. Specifically, when PKA phosphorylates CaMKK1 at Ser458, it prevents binding to Ca^2+^-CaM, thereby inhibiting CaMKK1 from responding to increased intracellular calcium levels [75,127]. Moreover, the principal site of autophosphorylation for CaMKK2 is Thr482, which generates a partial autonomous activity independent of Ca^2+^-CaM, disrupting the autoinhibitory mechanism of CaMKK2. Since CaMKK2 is not dependent on rapid fluxes of intracellular calcium for basal activity, it can respond to other stimuli for longer durations [7,74,75]. The temporal sequence of these two distinct signalling events, PKA phosphorylation on CaMKK and its ability to block Ca^2+^-CaM binding, plays an important role in directing signalling through CaMKK1.

A common characteristic of both CaMKKs is that they phosphorylate CaMKI and CaMKIV at residues Thr177 and Thr196, respectively [39,74,75]. In addition, both CAMKKs activate AMP-activated protein kinase (AMPK), a key regulator of cellular energy balance [128] (Figure 4). Sallé-Lefort et al. showed that the CaMKK/AMPK complex is involved in the activation of hypoxia-inducible factor-1α (HIF-1α). HIF-1α controls metastasis-associated lung adenocarcinoma transcript 1 (Malat1), enhancing its transcription upon low oxygen conditions. Their findings suggest that Malat1 expression is upregulated under hypoxic conditions by the CaMKK/AMPK/HIF-1α axis [129]. Another downstream substrate of CaMKKs is AKT (also known as protein kinase B or PKB), an important oncology target. Activated AKT promotes phosphorylation of cellular substrates, e.g., TSC1/2, FOXO, GSK3, and mdm2 [130], thereby controlling cell proliferation, metabolism, cell growth, and survival [75,131] (Figure 3).

Below we discuss the roles of CaMKK1, CaMKK2, CaMKI, and CaMKIV in CVD in more detail.

##### CAMKK1

The main role of CaMKK1 is phosphorylation of downstream CaMKI and CaMKIV, whereas the activation of CaMKK1 fully depends on CaM [39,74,75,126]. Recently, the crystal structure of CaMKK1 was revealed using two ATP-competitive inhibitors. The structural differences between CaMKK1 and CaMKK2 therefore can result in the generation of CaMKK-specific inhibitors [126]. Research supports the importance of CaMKKs in cardiovascular biology [126,132] via mTOR (mechanistic target of rapamycin) [133]. The mTOR pathway plays a key regulatory function in cardiovascular physiology and pathology (Figure 4) [134]. More recently it was shown that overexpression of CaMKK1 regulates the mesenchymal stem cell (MSC) secretome [135]. The overexpression of CAMKK1 in either MSC or upon direct injection of its encoding DNA into infarcted tissue resulted in improved cardiac function and increased vasculogenesis. In rats, injections in the heart of conditioned media from MSCs overexpressing CaMKK1 showed improvement in cardiac function after acute myocardial infarction, with increased vascular density and decreased scar tissue [135]. Moreover, the direct overexpression of CaMKK1 in infarcted tissue using a CaMKK1-encoding plasmid also significantly improved ejection fraction and decreased infarct size after acute myocardial infarction. Although the precise mechanism by which CAMKK1 affects regeneration is not known, these data put forward a novel role of CaMKK1 derived from the MSC secretome, indicating a potential therapeutic target for infarcted heart tissue [135].

Recently, the genetic variant rs7214723 of CaMKK1 was shown to be associated with a higher risk of developing CVD [136]. This genetic variant, which was shown to be associated with increased lung cancer risk [137,138], is a single-nucleotide polymorphism (SNP) that causes an amino acid change from glutamic acid (E) to glycine (G) at position 375 inside the catalytic domain of CaMKK1. This variation creates a charge change influencing substrate specificity, thereby inhibiting downstream CaMKI and CaMKIV [39,137,138]. This polymorphism appears to be highly represented in the population (MAF index: C = 0.3954 (1980/5008 1000 Genome)) [139]. Results from a cross-sectional study conducted on 300 cardiac patients showed a statistical difference between cardiopathic patients and European reference populations between the genotype and allele frequencies for rs7214723. These data suggest a potential role of rs7214723 that could be used as genetic biomarker in predicting CVD susceptibility [136].

Whilst interest in the function of CaMKK1 in the heart is growing, there is still lack of information regarding the role of CaMKK1 in vasculature and related vascular diseases.

##### CAMKK2

More research has been published on CAMKK2 compared to CAMKK1. Specifically, the interaction between CaMKK2 and AMP-activated protein kinase (AMPK) is well studied as it regulates many physiological processes such as hypothalamic control of feeding behaviour, hepatic gluconeogenesis, adipocyte differentiation and macro-autophagy [128,131,140]. AMPK is involved in a variety of cellular processes, including sensing intracellular ATP levels, regulation of autophagy and mitochondrial fission [141]. Since AMPK activity depends on CaMKK2, CaMKK2 has a possible role in cardiac energy production to protect the heart against calcium overload induced by sustained pressure loads [38]. Moreover, AMPK signalling is repressed during cardiac hypertrophy, and cardiac-specific knockout of AMPK promotes stress-induced cardiac hypertrophy [38,142]. Thus, the inactivation of CaMKK2 might indirectly result in the development of metabolic dysfunction and cardiac hypertrophy (Figure 4). Histone demethylase JMJD1C, an important epigenetic factor, represses the activation of AMPK during cardiac hypertrophy through the reduction of CaMKK2 expression. This confirms that AMPK signalling is involved in JMJD1C-mediated cardiac hypertrophy [143]. Absence of functional CaMKK2 in mice resulted in increased left ventricular dilatation and dysfunction and subsequent mortality [144]. This demonstrates that CaMKK2 can exert beneficial effects against pressure-overload-induced heart failure, thereby providing a therapeutic target for treatment of heart failure [144]. Additionally, CaMKK2 plays an important role in GLUT4 translocation through AMPK activation in cardiomyocytes. GLUT4, after metabolic stress, translocates from intracellular vesicles to the plasma membrane protecting cardiac myocytes from oxidative stress and ischaemic injury. During ischaemia, GLUT4 translocation represents a peculiar mechanism by which the heart increases glucose uptake increasing cell energy status. The pan-CaMKK inhibitor STO-609 as well as overexpression of a dominant-negative form of CaMKK2 in cardiomyocytes inhibited hydrogen-peroxide-mediated translocation of GLUT4 [143,144,145] (Figure 4). Further, CaMKK2 is involved in CaMKK2-AMPK-VASP/MLC signalling for migration of cells and for assembly of contractile actin stress fibres [146]. Additionally, it was shown that the regulation of mechanosensing is promoted by myosin-18 [135]. Exerted cellular forces are transduced via the cytoskeleton and are therefore of key importance in mechanosensing. Stress fibres are composed of actin (microfilaments) and non-muscle myosin II. Myosin-18B co-localizes with myosin II motor domains in these stress fibres. Moreover, myosin-18B knockout cells displayed the ability to exert forces to the environment, hence their involvement and importance in mechanotransduction [147,148].

Diabetes is an important risk factor for the development of CVD. In type 2 diabetic mice, endocrine hormone fibroblast growth factor 21 (FGF21) activates CaMKK2/AMPKα, thereby suppressing oxidative stress and enhancing endothelial nitric oxide synthase (eNOS) signalling, improving vessel relaxation [149]. This suggests the possible use of FGF21 to activate CaMKK2/AMPKα for therapeutic use of endothelial dysfunction in diabetes. Additionally, store-operated calcium entry (SOCE) is a central mechanism in cellular calcium signalling and in maintaining cellular calcium balance. Intracellular Ca^2+^ flux, driven by release of Ca^2+^ from intracellular stores can activate CaMKK2 via Ca^2+^/CaM [150]. This signalling pathway of SOCE-Ca^2+^/CaM-CAMKK2 is important for the regulation of autophagy and in maintaining proliferation and promoting the survival capability of endothelial progenitor cells (EPCs) [151]. This pathway is active at basal levels in cells of cardiovascular origin as an important homeostatic mechanism [152]. CaMKK2 activation and mTOR deactivation is associated with autophagy modulation (Figure 4). Here, a novel signalling pathway of SOCE-CAMKK2 in the regulation of autophagy was identified, revealing new insights in maintaining proliferation and survival capability of endothelial progenitor cells (EPCs). This offers ways to improve EPCs transplantation efficacy to enhance vascular re-endothelialization in patients with hypercholesterolaemia [151]. Moreover, tetrahydrobiopterin (BH4), a multifunctional cofactor implicated in regulation of nervous, immune, and cardiovascular systems, is a new potential endogenous activator of CaMKK2. BH4 targets CaMKK2 and promotes recovery of mitochondria in diabetic cardiomyopathy [153].

AMPK, among others downstream to CaMKK2, is a master sensor of cellular energy status involved in the progression of vascular calcification [154]. It is known that AMPK activators are associated with reduced calcification deposits [154], indicating a potential therapeutic role of AMPK in vascular calcification [155]. Vascular calcification is in part driven by the trans-differentiation of VSMC into cells with osteoblast characteristics, such as increased alkaline phosphatase activity and collagen II deposition, governed by the upregulation of Runt-related transcription factor 2 [156]. A recent study in mice showed that exogenous omentin-1 attenuates osteogenic differentiation of VSMCs through the activation of AMPK/Akt signalling [157]. On the contrary, the inhibition of AMPK and Akt signalling reverses the anti-calcific effect induced by omentin-1 both in vitro and in vivo [157]. Moreover, Lai and colleagues demonstrated that during calcification KMUP-3 (the xanthine derivative 7-[2-[4-(4-nitrobenzene)-piperazinyl]ethyl]−1,3-dimethylxanthine) inhibits both mTOR, downstream to CaMKK2, and β-catenin upregulation, essential for VSMC osteogenic phenotypic switching as well as enhancing AMPK activation inhibiting VSMC osteogenic differentiation [158].

#### 3.2.2. CaMKI and CaMKIV

Ca^2+^-CaM dependent kinase I and IV (CaMKI and CaMKIV) share the same common kinase structure (Figure 3), and their activation requires both binding of Ca^2+^-CaM and subsequent phosphorylation by Ca^2+^-CaM kinase proteins (CaMKKs) at Thr 177 and Thr 196, respectively [7,74,75]. The difference in activation between CaMKI and CaMKIV is that CaMKI remains entirely Ca^2+^-CaM-dependent, whereas CaMKIV can undergo an intra-subunit autophosphorylation of the Ser/Thr-rich N-terminus, generating Ca^2+^-CaM independent activity. This allows CaMKIV to maintain its functionality beyond increased intracellular calcium [39,74]. CaMKI is a monomeric kinase that is expressed by three different genes encoding α, β, and γ isoforms [159,160]. CaMKI is a cytosolic protein, which serves several functions including transcription activator activity, cell cycle control, hormone production, cell differentiation, actin filament organization, and neurite outgrowth (Figure 4). However, the precise role of CaMKI in CVD is not completely understood [39].

CaMKIV is a monomeric kinase and can be expressed as two isoforms (α and β), encoded by a single gene [161]. CaMKIV is considered important in the nervous system [162,163], but recently it received attention for its role in cardiovascular pathophysiology [164]. Several studies showed an association of the genetic variant rs10491334 in CaMKIV with elevated diastolic blood pressure [162,163,164] (Figure 4). Furthermore, rs10491334 was also associated with a reduction in the expression level of CaMKIV in hypertensive patients [164]. Moreover, CaMKK/CaMKIV has been shown to be a key endogenous protective pathway in ischaemia and an important regulator of blood–brain barrier integrity [165]. The deletion of CaMKK or CaMKIV in mice exacerbates stroke outcome, such as infarct volume, oedema formation and behavioural deficits. Liu et al. described furthermore that the CaMKK pathway is involved in the immune response to brain injury, increased blood brain barrier impairment, transcriptional inactivation of cAMP, and inflammatory responses in females after stroke. The CaMKK signalling pathway might therefore be a potential target for stroke treatment [166].

In the vasculature, it has been shown that the interaction between CaMKKs and CaMKIV is important in the regulation of CREB during phenotype switching of VSMCs [34] (Figure 4). Indeed, nuclear CaMKIV plays a crucial role in the activation of CREB, whereas CaMKII inhibits CREB by phosphorylating it [167,168]. Previous studies demonstrated that both CaMKII and CaMKIV can phosphorylate Ser^133^ of CREB. However, CaMKII also phosphorylates Ser^142^ in the transcriptional activation domain of CREB. This event has been reported to inhibit CREB activity by interfering with CREB dimerization and protein interactions to form an active promoter complex [169,170]. It has been shown that VSMC phenotype depends on the activity and form of different calcium-dependent transcription factors, such as CREB [34,167]. Part of the VSMC phenotype regulation is therefore under the control of the Ca^2+^-CaM kinase pathway, thus making this an interesting therapeutic option to regulate VSMC phenotype switching.

## 4. Conclusions

In this review, we have summarized the current state of the literature on Ca^2+^-CaM -dependent kinase cascade functions, with a focus on the heart and vessels (Table 2). Our review clearly points towards an important role for calmodulin and calcium kinase proteins in CVD. CaMKKs are involved in regulating calcium signalling in cardiovascular metabolism resulting in calcium-induced autophagy, an important homeostatic mechanism active at basal levels in cells of cardiovascular origin [151,152]. Moreover, the CaMKK/CaMKIV phosphorylation cascade has been shown to be a key endogenous protective mechanism in ischemia and an important regulator of blood–brain barrier integrity [165]. In addition, genetic variants in Ca^2+^-CaM dependent kinases were associated with CVD [39,136,162,164]. Further, it is important to underline the role of Ca^2+^-CaM-dependent kinases in VSMC integrity. Whilst it is known that the interaction CaMKK2/CaMKIV is involved in phenotype switching of VSMCs [34], data are lacking as to the role of CaM and CAMKK1 in vessel wall homeostasis. As the Ca^2+^-CaM-dependent kinase cascade regulates many cellular processes in both cardiac and vascular tissue, further investigation of this signalling pathway is necessary to understand this process and to open up novel avenues for the treatment of CVD.

## Figures and Tables

**Figure 1 ijms-23-16139-f001:**
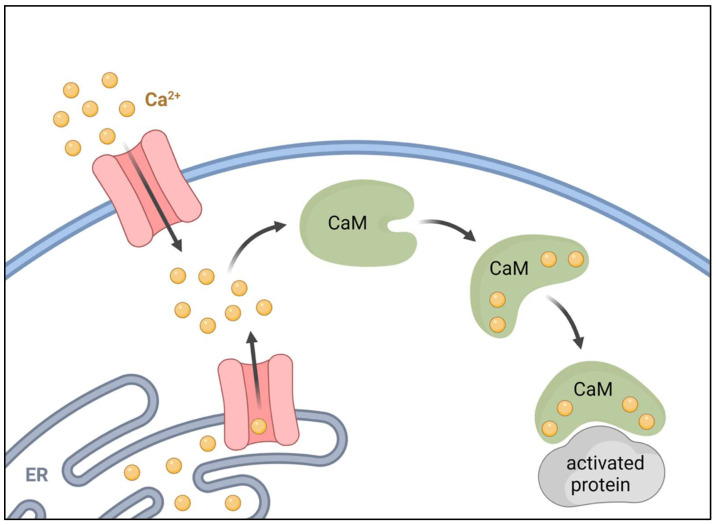
Schematic representation of Ca^2+^-CaM activation. After binding 4 ions of Ca^2+^, calmodulin undergoes conformational change that leads to activation of downstream proteins.

**Figure 2 ijms-23-16139-f002:**
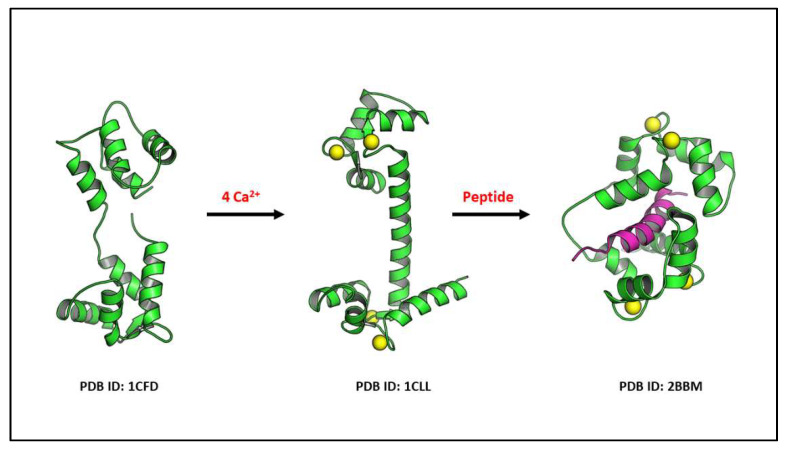
The 3D illustrations show the flexibility of calmodulin in binding to 4 Ca^2+^ and undergoing conformational changes for its reaction with a specific downstream protein, such as the CaM kinase proteins. Specifically, the binding of the calcium–calmodulin complex to auto-inhibitory target protein regions leads this conformational change and the target protein activation. Ca^2+^ is displayed as yellow balls and peptide is shown as magenta ribbon.

**Figure 3 ijms-23-16139-f003:**
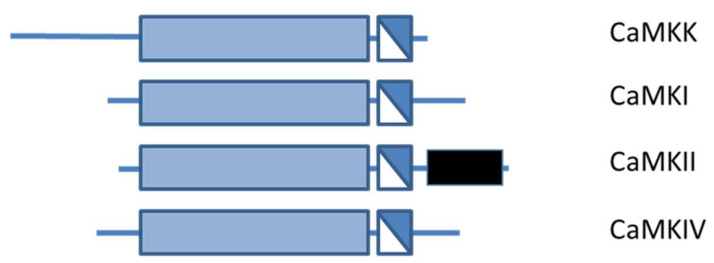
The general structure shared by the CaM-kinases. The N-terminal catalytic domain is represented in light blue, followed by a regulatory region containing in white the autoinhibitory domain and in dark blue the CaM-binding domain. The black block shows the C-terminal association domain of CaMKII. Adapted from Beghi S, et al., 2020 [39].

**Figure 4 ijms-23-16139-f004:**
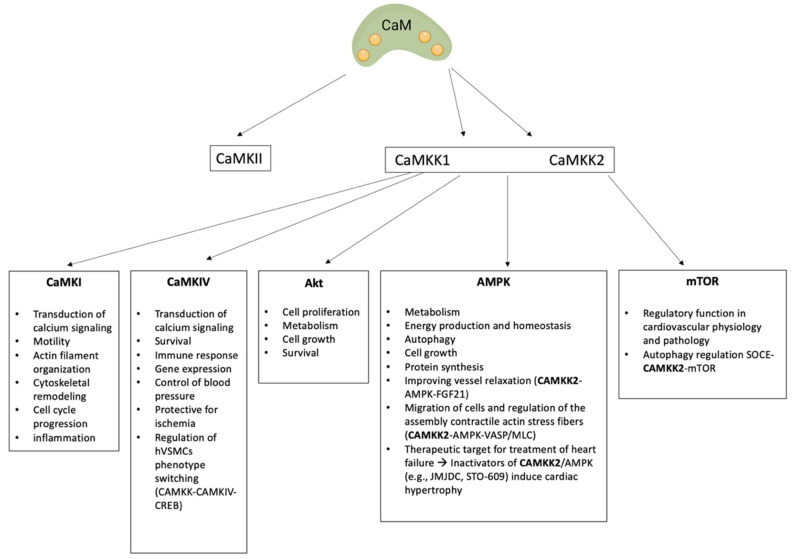
Ca^2+^-CaM kinases downstream of calmodulin. Binding of calcium to calmodulin leads to activation of the Ca^2+^-CaM-dependent kinase cascade, consisting of CaMKK1 and CaMKK2, CaMKI, and CaMKIV, as well as activation of CaMKII. The activation of CaMKKs regulates crucial cellular functions. Calcium calmodulin mediates downstream signalling of CaMKI, CaMKIV, AMPK, Akt, and mTOR.

**Table 1 ijms-23-16139-t001:** List of the main human CaMs gene polymorphisms. CaM polymorphisms are associated with a spectrum of arrhythmic phenotypes. Adapted from Kotta et al. [61].

Gene	Nucleotide Change	Amino Acid Change	Associated Phenotype	Refs.
*CALM1*	c.389A>G	p.D130G	LQTS	[59,61]
*CALM1*	c.426C>G	p.F142L	LQTS	[59,61]
*CALM1*	c.161A>T	p.N54I	CPVT	[61,66]
*CALM1*	c.293A>G	p.N98S	LQTS, CPVT	[61,66]
*CALM1*	c.268T>C	p.F90L	IVF, SUD	[67,69]
*CALM1*	c.395A>T	p.D132V	LQTS	[64]
*CALM2*	c.293A>G	p.N98S	LQTS, SUD	[61,62,68]
*CALM2*	c.287A>T	p.D96V	LQTS	[59,61]
*CALM2*	c.293A>T	p.N98I	LQTS	[61,62]
*CALM2*	c.400G>C	p.D134H	LQTS	[38,39]
*CALM2*	c.389A>G	p.D130G	LQTS	[61]
*CALM2*	c.396T>G	p.D132E	LQTS, CPVT	[61,62]
*CALM2*	c.394G>C	p.D132H	LQTS	[64]
*CALM2*	c.407A>C	p.Q136P	LQTS, CPVT	[62]
*CALM3*	c.389A>G	p.D130G	LQTS	[61,63]
*CALM3*	c.308C>T	p.A103V	CPVT	[61]
*CALM3*	c.286G>C	p.D96H	LQTS	[61,65]
*CALM3*	c.426T>G	p.F142L	LQTS	[61,65]

**Table 2 ijms-23-16139-t002:** Summary of the involvement of the CaM cascade in cardiovascular disease.

Kinase	In Heart Pathology	In Vasculature Pathology
*CaM*	Polymorphisms linked to calmodulinopathy, arrhythmia [44,45,59,60,61,62,63,64,65,66,67,68,69].	Unknown
*CaMKII*	Implicated in myocardial injury [116], atrial fibrillation [117], cardiac hypertrophy, ischaemia/reperfusion injury [118], heart failures, contributing to apoptosis, arrhythmias [113], defective ECC and ETC [53,114,115], pathological hypertrophy [96] and contractile dysfunction during heart failure [54,96,114,115].	Regulation of VSMCs phenotype switching through the inhibition of CREB [34].
*CaMKK1*	Polymorphism linked to the higher risk to develop CVD [136].	Regulation of VSMCs phenotype switching (CaMKK-CaMKIV-CREB) [34].
*CaMKK2*	Inactivation of CaMKK2 indirectly results in the development of metabolic dysfunction and cardiac hypertrophy [143].	Regulation of VSMCs phenotype switching (CaMKK-CaMKIV-CREB) [34].
*CAMKI*	Unknown	Unknown
*CAMKIV*	Polymorphisms linked to elevated diastolic blood pressure [162,163,164].	Regulation of VSMCs phenotype switching (CaMKK-CaMKIV-CREB) [34].

## Data Availability

Data are available upon reasonable request from the corresponding author.

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
