# Peer review of "Calcium Signalling in Heart and Vessels: Role of Calmodulin and Downstream Calmodulin-Dependent Protein Kinases"

_ijms, 2022, doi:10.3390/ijms232416139_

Round 1
Reviewer 1 Report
The review is nice, covering a lot of literature and covers an interesting research topic. Figures can

Author Response
Point 1: Figure 1 seems trimmed and incomplete.
Answer 1: We thank the reviewer for looking critically at our figures. Figure 1 is complete, we adjusted the frame to delineate the figure better.
Point 2: Adding some recent literature might aid in the improvement of some points https://doi.org/10.2174/1381612825666190312115140
Answer 2: We agree with the reviewers suggestion. We added additional recent literature.
Point 3: Some more figures need to be added to make it more attractive.
Answer 3: Although we agree with the reviewer that figures make more attractive the review, but considering the complexity of the argument and the lack of information for some argoument, we preffered to show simple and schematic figures in order to sum up the concept exposed in the review.
Point 4: English language needs to be proofread as some sentences are long and need restructured.
Answer 4: We have read our manuscript again as well as let it proof-read by a native English speaking person.
Reviewer 2 Report
Reviewer’s comments to « Calcium signalling in heart and vessels - Role of calmodulin and downstream dependent protein kinases” by Beghi et al. (ijms-1988912)
In this work Authors review the role of calmodulin and calcium-calmodulin dependent protein kinases of the CaM-kinase type in the context of cardiovascular biology. After a brief general introduction on calmodulin, CaM-kinases and cardiovascular physiopathology, Authors discuss the importance of calcium-dependent signal transduction in cardiomyocytes and vascular smooth muscle and endothelial cells. A special section is included on basic biochemistry and genetics of calmodulin, and on the biochemical mechanisms of calcium-calmodulin regulated activation of calmodulin target proteins and enzymes, and calmodulinopathies are presented. After a general introduction of CaM-kinases, these enzymes are discussed according to the kinase isoform (CaMKK1, CaMKK2, CaMK1, CaMK4) and CaMK-CaMKK regulation is discussed, and pathophysiological implications are mentioned.
Considering the importance of calcium signaling in muscle contraction and the regulation of endothelial cell biology, this work may be of interest for readers who wish to familiarize with the involvement of calmodulin and CaM-kinase enzymes in cardiovascular pathophysiology.
Some modifications are required for clarity, as detailed below.
Comments:
Lanes 2-3 (Title) : “Calcium signalling in heart and vessels
Role of calmodulin and downstream dependent protein kinases” :
Calcium signalling in heart and vessels - Role of calmodulin and downstream calmodulin-dependent (or calmodulin-dependent downstream) protein kinases
Lane 4: “MalgorzataFurmanik” : Malgorzata Furmanik (and please homogenize spaces, commas and numbering in the list of the Authors.)
Lane 10: “Maastricht University Medical Centre+” : +?
Lanes 34 and 41 : There is some confusion between “disease” and “diseases”: from a pathophysiological standpoint, cardiovascular diseases are clearly several distinct entities, not a single disease.
This is reflected in lane 41, because if : “Among…”, then these are diseases, not a single disease.
Lane 25: “the calcium calmodulin kinase family that are involved in the regulation of…” : …that is involved in…
Lanes 42-43: “(atherosclerosis)…can block blood flow to the heart completely” : the Reviewer is not sure 1) whether during atherosclerosis blood flow to the whole heart is usually blocked completely, and 2) whether atherosclerosis can be considered as an “event”. Please use the notions of atherosclerosis and ischemic stroke more precisely.
Lanes 44-45: “regulation of calcium signaling play a crucial role” : “plays”
Lanes 48-50: “CaM ensures activation of downstream calcium calmodulin kinase proteins (CaMKs), including CaMKK1, CAMKK2, CaMK1, CaMK2 and CaMK4. CaMKs are multifunctional proteins and some are substrate-specific kinases.” : “multifunctional” is somewhat problematic here, as the multi-substrate kinases still accomplish a single function (i.e.: ser/thr phosphorylation). “Multifunctional” implies other functions in addition to ser/thr phosphorylation (even if the ser/thr phosphorylation occurs on multiple substrates.)
Lanes 54-61 : The process of calcification requires more detailed explanation. For non-specialist readers it would be useful to briefly discuss, whether this is an intra- or extracellular process, what are the mechanisms of calcium deposition, and how are these related to intracellular calcium signaling in live cells, in which free calcium levels are low overall. If “calcification is an active process” (lane 56), then the mechanisms of this process need to be briefly discussed (idem for aortic aneurism and aortic valve stenosis, lanes 54 and 60.)
Lane 67: “Calcium is an abundant element in nature and the most abundant mineral in the human body.” Please compare this to other metals (K, Na, Mg).
Lanes 72-73 : “The calcium concentration outside the cell is some 2mM, thus a factor of 20,000 higher.” Please briefly discuss active calcium transport here (see also lanes 77-78 and 118.)
Lanes 74-75: “The presence of calcium-sensing receptors, such as calmodulin, is essential to sense the calcium ions in the extracellular fluid and to activate a series of downstream pathways that directly regulate cellular functions.” In the opinion of the Reviewer, as well as of the Authors (see lane 123), calmodulin is actually intracellular. Sensing of extracellular calcium is accomplished by calcium sensing receptor, a very different, G protein coupled receptor, also called CaSR or GPRC2A, located in the plasma membrane. If Authors really wish to elaborate on the notion of calmodulin sensing extracellular calcium, this must be substantiated by extensive bibliographical references. It is also not clear, how a soluble protein such as calmodulin, if located in the extracellular space, could signal towards the interior of the cell.
Lanes 77-78: “…spaces between intracellular organelles and delimited compartments, as sarcoplasmic reticulum.” : “…such as the sarcoplasmic reticulum…”
Lanes 81-86: Please include additional References, for example for the role of calcium in blood coagulation, platelet aggregation, as well as in gene expression regulation.
Moreover, please use Arabic or Latin numerals for the various CaMK enzymes consistently in the text, and also in Fig. 2.
Lanes 85-86: “such as hypertrophy” : such as cardiac hypertrophy.
Lanes 87-102: MLCK biology and its regulation by calcium, as well as the role of calcium in smooth (and cardiac) muscle contractility and the distinction between contractile and “synthetic” vascular smooth muscle cells, as well as known inducers of such phenotypic changes need to be presented more clearly and in greater detail. Calcium entry via extracellular vesicles, and the mechanisms of vessel wall calcification also need to be discussed more clearly and in a more detailed fashion.
Lanes 101-102, 201 and 245-250: The role of calcium oscillations in the regulation of CaMK enzymes is mentioned in several instances in the Manuscript. However, in the opinion of the Reviewer, this is not done in a sufficiently structured and precise fashion, in order to permit to non-expert readers to understand the corresponding biochemical mechanisms, such as the acquisition of calcium-independent kinase activity and its implications for signal transduction. Calcium regulation of CaMK2 and 4 is very interesting and complex. Maybe a corresponding Figure could also be helpful.
Also, in general, the Reviewer has the impression that the various Figures of the present Manuscript are too simplistic, and that amelioration of the Figures could greatly enhance the quality of the Paper. Figures should be more detailed and should convey more structural and mechanistic information. Improvement of the quality of Figures would greatly enhance the quality of the Paper and would be very helpful to the readers.
Lane 113: “Due to the cooperativity of the intra and inter lobe, CaM has evolved a high sensitivity to changes in intracellular calcium levels.” : this phrase needs to be reformulated; in the opinion of the Reviewer, the expression “due to” is not appropriate in this context. (Of course, the Reviewer agrees with the idea that the evolution of the structure of calmodulin was very probably greatly influenced by the possibility of acquiring a high sensitivity for calcium, and that this occurred through the “discovery”, by Darwinian evolution, of the structure allowing high cooperativity (and other features that allow interaction with downstream protein partners also evolved.)
Also, please note that “inter lobe” is a somewhat confusing term, because the interlobular region is not a lobe (a Figure may be helpful here to show an EF-hand and CaM conformational changes and exposure of interacting surfaces on the molecule and CaM-partner binding and activation.) The notion of cooperativity of calcium binding needs to be explained as well.
Lane 111: It would be nice, if Authors could discuss the eventual advantages, for the organism, of the existence of three distinct calmodulin genes that code for an identical protein.
Lanes 120-122: “These proteins have been shown to play important roles in maintenance of plasma membrane integrity and cytoskeletal structure, postsynaptic function, and in the regulation of smooth muscle cell contraction, respectively. Please add here relevant References for the function of spectrin, neuromodulin, caldesmon etc.
Lanes 123-124: “a multitude of process…” : processes (idem lane 204.)
Lane 124: Why “subcellular”? In which sense, for example, differentiation is a process that has a subcellular localization? Please reformulate.
Lanes 125-127: “calmodulin undergoes a conformational changes that lead to activation of downstream proteins.” : calmodulin undergoes a conformational change (or: undergoes conformational changes) that leads (or: that lead) to activation of downstream proteins.
Lanes 129-137: Please extend, give some more mechanistic detail, and illustrate.
Lane 132: “voltage-gated calcium (Ca2+)channels” : voltage-gated calcium (Ca2+) channels.
Is “(Ca2+)” necessary here?
Lanes 146-147: Is this related to sudden infant death syndrome (SIDS)?
Lanes 171-172: “are associated with different type of arrhythmic phenotypes” : “with different types of…” or “with a different type of…”
There is some uncertainty in this Paragraph and the corresponding Table, due to the fact that three different calmodulin genes code for an identical protein. It would be nice, if Authors could discuss in more detail, for example, whether a given calmodulin gene is specifically associated with a specific pathophysiology. For example, p.D130G with LQTS is found in all three genes (and this corresponds to the same c.389A>G, but, for example, p.F142L is absent in CALM2. In other words, are the three genes equivalent, and whether the association of a specific calmodulin gene with a calmodulinopathy is due simply to the fact that the corresponding mutation happened to occur in that specific gene (and not in another) “by chance”?
Also, do the three calmodulin genes have gene-specific non-coding sequences? Are these known to be involved in physiopathology? Why are there three genes for the same protein? Do other examples exist for this type of evolution? Some guidance for non-experts would be very helpful.
In addition, please briefly discuss, what is known regarding the molecular physiological basis of these calmodulinopathies.
Lanes 176-178: Please give some mechanistic/molecular detail for the calcium-calmodulin dependent regulation of contractility.
Lane 181: “drivenvascular”?
Lane 181: “The role of CaM in VSMC-driven vascular disease (i.e. atherosclerosis, aneurysm, calcification)…” the Reviewer is not entirely sure that the role of, for example, endothelial cells can be excluded with confidence, in the ætiology of some of these pathologies.
Lane 185: “amplification of intracellular calcium” ? …intracellular calcium signals…?
Lane 189: “protein bound Ser or Thr”: the Reviewer believes that something like “Ser or Thr residues in proteins” would be more appropriate. “Protein-bound” is more adequate in the context, for example, of coenzymes or prosthetic groups.
What determines which ser/thr residues will be phosphorylated?
Somewhere around lane 190 it would be nice to explain what CaMKK enzymes are, otherwise it may be difficult to apprehend why CaMKK is a CaMK. The concept that some calmodulin-activated kinases (CaMKK) may phosphorylate other calmodulin activated kinases (CaMK) is not self-evident for the non-initiated reader. A Figure for the cascade and downstream molecular targets, as well as for the sterical mechanism of CaM-induced activation would be helpful.
Lanes 191-192: “The second group consists of CaMKIII, phosphorylase kinase and myosin light chain kinase and are (which are) substrate-specific kinases with only one downstream target.” : The Reviewer believes that, considering the title of this Paper, it would be very appropriate to devote a specific paragraph (and Figure) also to this family of CaM-activable enzymes.
Lane 201: “can be independent of CaM…” : “can become independent of CaM…” : please discuss the underlying mechanisms.
Fig. 2: Downstream targets of CaMK2 should also be indicated (with corresponding arrows). Also, it is not clear in this Figure, whether CaMK1 and 4 are activated exclusively through CaMKK1 or whether they can be activated directly by CaM.
Lane 228: “membrane excitement”? Please elaborate.
Lanes 237-238: “The effect of CaMKII on ECC is crucial because it phosphorylates calcium handling proteins including sarcoplasmic reticulum (SR) calcium release channels, ryanodine receptors…” “such as ryanodine receptors”?
Lanes 241-243: “The phosphorylation of myofilament proteins (cardiac myosin binding protein-C) and myosin regulatory light chain 2 by CaMKII contributes to myofilament calcium sensitization and interactions”. Please give some mechanistic information.
Lanes 243-244; “CaMKII targets cardiac Na+ and K+ channels, which are involved in the regulation of calcium homeostasis” Please elaborate (activation? inhibition? mechanisms?)
Lane 252: “The altered hyperactivate regulation of” : ? The altered, hyperactivated regulation of (or the hyperactivation of?)
Lane 266: “s-adrenergic” : ?
Lanes 267-268: “play critical roles in … human pulmonary VSMC calcification under hypoxic conditions” : would it be possible to mention corresponding human physiopathological conditions here?
Lanes 281-283: “The main difference between both CaMKKs is that CaMKK1 is kept in an activate (?) state until binding of Ca2+-CaM relieves the autoinhibitory mechanism, while CaMKK2 has partially autonomous activity in the absence of Ca2+-CaM.” : This phrase is not clear. Please clarify.
Lanes 292-293: “The temporal sequence of these two distinct signalling events, PKA phosphorylation on CaMKK and its ability to block Ca2+-CaM binding, plays an important role in CaMKK-dependent signalling.” Please enhance clarity regarding the relationship between the temporal PKA- and CaM-dependent regulation of CaMKK-dependent signalling and also regarding its importance.
Lanes 311-312: “Activated AKT promotes phosphorylation of cellular substrates, thereby inducing cell proliferation, metabolism, cell growth and survival” : please specify AKT substrates.
Lanes 316-317: “The main role of CaMKK1 is phosphorylation of downstream CaMKI and CaMKIV, in which activation of CaMKK1 fully depends on CaM” : it is not clear to the Reviewer, what “in which” means in this context. Also, please enhance clarity regarding differences regarding the autonomous activity (or lack thereof) of the various enzymes.
Lanes 325-326: “decreased scare tissue.” : scar tissue.
Lanes 322-329: This section is not clear. “CaMKK1 is involved in the regulation of the mesenchymal stem cell (MSC) secretome” : so CaMKK1 in MSC regulates what will be secreted by MSC.
And: “In rats, injections in the heart of conditioned media from MSCs overexpression CaMKK1 showed improvement in cardiac function” : “overexpressing CaMKK1”?
And : “The direct overexpression of CaMKK1 in infarcted tissue using a CaMKK1-encoding plasmid significantly improved ejection fraction and decreased infarct size after acute myocardial infarction.” Here CaMKK1 is expressed in heart muscle, not in MSCs. And: “These data put forward a novel role of CaMKK1 derived from the MSC secretome, indicating a potential therapeutic target for infarcted heart tissue.”: is this supposed to mean that CaMKK1 may be part of the MSC secretome (i.e.: derived from it/secreted by MSC?) Lanes 322-329 need to be rewritten in depth.
Lane 348: “Since AMPK activity depends on CaMKK2” : please give full name of AMPK, and briefly explain its biochemical function.
Lanes 356-357: “Inhibition of CaMKK2 in mice resulted in increased left ventricular dilatation and dysfunction and subsequently mortality.” Please add References.
Lane 361: “overexpression of the dominant-negative form of…” : of a dominant-negative…?
Lane 362 : Please explain the relevance of GLUT4 translocation.
Lanes 364-365: “Additionally, it was shown that regulating mechanosensing is promoted by myosin-18.” : it would be nice, if Authors could state here, how does this relate to CaM-kinases.
Lane 370: Please briefly explain SOCE.
Lane 375: Please add References for autophagy and SOCE-CaMKK2 here.
Lanes 381-382: “AMPK, downstream to CaMKK2, is a master sensor of cellular energy status involved in progression of vascular calcification” : it should be noted that AMPK is downstream not only from CaMKK2.
Lane 386 : calcific? Also, the trans-differentiation of VSMC to osteoblasts requires some more detailed discussion.
Lane 388: “(xanthine derivative 7-[2-[4-(4-nitrobenzene)-piperazinyl]ethyl]−1,3-dimethylxanthine)” : (the xanthine derivative, 7-[2-[4-(4-nitrobenzene)-piperazinyl]ethyl]−1,3-dimethylxanthine).
Lane 390 : the notion of “phenotypic switching” needs clarification : is this a switch between a proliferative and a “synthetic” phenotype (as used in lanes 263-264), or is this related to a shift towards an osteoblastic phenotype? (Please see also lanes 416 and 434.) If “phenotype switching is used to mean two different types of phenotypic changes, this needs to be done unambiguously.
Lane 405: “monomeric kinase consisting of two isoforms” : if monomeric, then it cannot consist of two isoforms (which would make a dimer). Please reformulate.
Lanes 413-414; “This data suggests that CaMKK signalling plays a protective role in stroke development.” Is this related to the development of stroke proper, or rather, to the evolution of the situation following a stroke (such as recovery?)
Lanes 416-417: “Indeed, (not “.”) nuclear CaMKIV plays a crucial role in the activation of CREB, whereas CaMKII inhibits CREB phosphorylation.” Please add functional consequences of CREB phosphorylation and inhibition thereof.
Lanes 420-421: “Part of the VSMC phenotype regulation is thus under the control of the Ca2+-CaM kinase pathway. Thus, making this an interesting therapeutic option to regulate VSMC phenotype switching.”: Part of the VSMC phenotype regulation is thus under the control of the Ca2+-CaM kinase pathway, thus making this an interesting therapeutic option to regulate VSMC phenotype switching.
Lanes 431-432: “genetic variants in Ca2+-CaM dependent kinases was associated with CVD” : were/are/have been shown to be associated…
It would be nice, if the visual format of Table 2 could be improved for greater ease of reading.
Please use a single format for Reference numbering in the text (see for example lanes 58 vs. 79).
Also, please note that in References, in many instances citations are incomplete: year, volume, page numbers are missing.
Lane 448 : what is “visualization” as author contribution?
Lanes 449-450 and 452-453: was funding Chinese and/or European?
Lane 494: “Matchkov V v.,” : v?
In conclusion, Authors are encouraged to improve the structure and the overall clarity of the text in general. Additional and more detailed Figures would also be helpful. Some of these may be added as accompanying material. The use of some complex notions requires a brief specific introduction, as it cannot be expected that readers are already familiar with them. Also, in the opinion of the Reviewer, auto-activation and calcium-independent activity of CaM kinases, as well as CaMK-CaMKK interactions or excitation-contraction coupling, CREB function, phenotype switches, etc., are mentioned at several instances in a somewhat diffuse or disjointed manner. A coherent and concise introduction for such notions before they are discussed in the context of cardiovascular molecular pathophysiology would facilitate the reading of this interesting and informative work.
Reviewer 3 Report
The article is devoted to understand the role of calmodulin and its downstream activation pathways of the calmodulin kinase family on the concentration of intracellular calcium.
When we talk about contemporary literature, we mean studies and articles of the last 5 years (the last 10 years maximum). I recommend trying to replace old sources with new ones. The references should be put in order. There are a lot of links in which information is cut off on the title of the journal and you need to manually search for an article to view it for relevance. For a more convenient search, I would recommend that authors indicate the «doi» of the article.
It would be appropriate to provide a link to the statistics of deaths from СVD in line 39.
Line 43. The authors should expand the last sentence with references. Depending on the location of the vessel blocked by the plaque, not only ischemic stroke can occur, but also coronary artery disease, peripheral arterial disease ...
Сhapter 3.1. CAMKII. Line 233. I would invite the authors to describe these «several cardiac functions». It would be nice to describe how isoforms CaMKII γ and δ are similar or different in function.
Сhapter 3.2.1.2 CAMKK2. Line 345. It would be great to add the thoughts of the authors here. Why do you think CAMKK2 is more explored than CAMKK1?
In conclusion, I want to say that the article is written in easy language, it was interesting to read it. The article can be recommended for publication after revision.
Round 2
Reviewer 2 Report
Several changes have been made in the original Manuscript following the comments of the Reviewer, and these improved the Paper. Some additional issues will require additional attention by the Authors, necessary for publication, as below. Essentially, clarity of specific parts of the text need to be significantly improved; this will enhance the quality of this work.
Lanes 27-28: “Among them, the calcium calmodulin kinase family that is involved in the regulation of cardiac functions.” : Among them is the calcium calmodulin kinase family that is involved in the regulation of cardiac functions.
Lanes 39-40: “Indeed, different factors coexist and cooperate together increasing the risk to initiate and progression of CVD.” : “to initiate and progression”?
Lanes 49-50 : “Among the different pathways involved in cardiovascular homeostasis, regulation of calcium signaling play a crucial role.” : plays
Lanes 56-68 : “is involved in many cellular processes, including glucose homeostasis, apoptosis, hematopoietic stem cell maintenance, normal immune cell function and cell proliferation.” Please add References.
Lane 62 : “activly” : actively
Lanes 76-77 : “The aim of our review is to summarize the current state-of-the-art on the role of calcium signalling, with a focus on calmodulin and its downstream calcium kinases.” …in CVD.
Lanes 86-87 : “The concentration of calcium within the cell is some 100nM.” : The concentration of calcium within the cytosol is some 100nM. (RE is, for example, much higher.)
Lanes 89-90 : “This calcium entry is regulated by calcium-sensing receptors (CaSR) which are essential to sense calcium ions in the extracellular fluid and facilitate active and orchestrated calcium uptake.” Please add an adequate Reference for this. In addition, please discuss here calcium channels and calcium pumps that are responsible for the calcium concentration gradients and fluxes observed in cells.
Lane 95 : “The main function of calcium in the heart is the regulation of excitation-contraction coupling (ECC).” : in the opinion of the Reviewer, calcium is an essential central player in the core function of ECC.
Lane 104 : “in the regulating blood…” : “in regulating blood…” or “in the regulation of blood…”
Lane 117 : “uptake into synthetic VSMCs causes enormous stress…” : What are synthetic VSMCs? Artifical VSMCs? And what is taken up into them? Please use “synthetic” correctly or reformulate (“VSMCs of the synthetic phenotype…”) or use an abbreviation such as, for example SVSMC for these). Lanes 113-124 need to be rewritten for enhanced clarity.
Explain “trapping mechanism”.
Lane 133 : “that encode for three identical proteins.” : an identical protein?
Lanes 134-135 : “that may permit regulation of CaM…” : what type of regulation? Regulation of expression? Regulation of localization?
Fig. 1 : Please include a 3D illustration regarding the main conformational change that occurs in calmodulin during calcium binding.
Fig. 1 : Calmodulin-induced activation of, for example, PMCA-type calcium pumps is discussed in lane 143. Please include a schematic diagram regarding the mechanism of activation by calmodulin (i.e.: binding of calcium-calmodulin to a calmodulin-binding autoinhibitory region in PMCA leading to abolished PMCA autoinhibition.) Or, alternatively, illustrate the mechanism of CaMK activation by calcium-calmodulin (as discussed in lanes 233-241.)
Lane 166 : “in the contest” : in the context; “it is important to consider…”; “that are functional antagonist” : antagonists
Lane 221 : “VSMC driven” : VSMC-driven
Lanes 242-244 : Discuss the mechanism of acquisition of calcium-independency of the enzymes in greater mechanistic detail. How does CaMK4 become calcium-independent?
Lanes 246-247 : “In the following sections we describe in more detail the role of CAMKs in heart and vasculature.” : In the following sections we describe in more detail the role of the multifunctional CAMKs in heart and vasculature.
Lanes 286-287 : “Moreover its phosphorilation increase the force…” : Moreover, its phosphorylation increases the force…
Lane 292 : “Additionally, CaMKII targets activate cardiac Na+ and K+ channels, which are involved in the regulation of calcium homeostasis, through phosphorylation.” : Additionally, CaMKII targets and activates, through phosphorylation, cardiac Na+ and K+ channels, which are involved in the regulation of calcium homeostasis.
Lanes 314-315 : “The rise of intracellular Ca2+ in VSMCs promotes nuclear translocation of Ca2+-CaMKII, which then activates CaMKIV phosphorylation of CREB.” : Please discuss the mechanism of CaMKII-induced CaMKIV activation here.
Lanes 378-379 : “More recently it was shown that CaMKK1 is involved in the regulation of the mesenchymal stem cell (MSC) secretome.” Is it the MSC secretome which is regulated by CaMKK1, or is it the MSC-secreted CaMKK1 that regulates cardiac function? Or is it CaMKK1 in MSCs that induces in them the secretion of unspecified substances that act on cardiac function? This section is still written is a somewhat confuse manner.
And how is this related to the direct transfection of CaMKK1 into infarcted heart tissue?
Is CaMKK1 secreted and then acts extracellularly? (“CaMKK1 derived from the MSC secretome”, lane 385)
Please reformulate and clarify.
Lane 403 : “AMP activated protein kinase” : AMP-activated protein kinase
Lanes 433-434 : “Excerted cellular forces are transduced via the cytoskeleton, and therefor of key importance in mechanosensing” : Exerted cellular forces are transduced via the cytoskeleton, and are therefore of key importance in mechanosensing
Lanes 442-444 : “Additionally, store-operated calcium entry (SOCE) is a central mechanism in cellular calcium signalling and in maintaining cellular calcium balance and linked to CaMKK2 is also involved in SOCE-CAMKK2-mTOR signalling, which is important for calcium-induced autophagy regulation.”
Please reformulate, as the phrase and the connection with SOCE are not clear.
The clarity of this section (lanes 438-454) has to be enhanced considerably.
Lanes 472-473 : “and subsequent phosphorylation of Ca2+-CaM kinase kinase proteins” : and subsequent phosphorylation by Ca2+-CaM kinase kinase proteins” ?
Lanes 498-499 : “Indeed, nuclear CaMKIV plays a crucial role in the activation of CREB whereas CaMKII inhibits CREB phosphorylation.” : Please add mechanistic detail for the inhibition of CREB phosphorylation by CaMKII. Explain, how does this kinase inhibit phosphorylation?
Clarity of this section (lanes 497-508) needs to be improved.
Lanes 537-541 : “Funding: This work was funded by the National Natural Science Foundation of China (No.61904070) and the Jiangxi University of Science and Technology Science Research Starting Foundation (205200100117).”
It is stated in the Response to Reviewer, that funding was European (answer N°71). Please explain why Chinese funding is then included in the revised Manuscript.
Lane 596: Ref.22: “Matchkov, V.v.” : Matchkov, V.V. (uppercase).
Round 3
Reviewer 2 Report
Lanes 104-105 : “These calcium channels include, amongst others, the plasma membrane calcium ATPase (PMCA) pump and the sodium/calcium ion exchanger (NCX)).” Please note, that PMCAs and sodium/calcium exchangers are not channels. And please delete second “)”.
Lanes 90-91 : “This calcium entry is regulated by calcium-sensing receptors (CaSR) which are essential to sense calcium ions in the extracellular fluid and facilitate active and orchestrated calcium uptake.” Please insert References here at the end of this phrase specifically for the calcium sensing receptor protein (CaSR) and its role in calcium entry in cardiovascular biology.
Lanes 123-150 : Text needs further clarifications. For example : “Synthetic VSMCs shed extracellular vesicles…” (lane 127) and “These extracellular vesicles form the nidus for VC and their uptake into synthetic VSMCs causes cellular stress…” (lanes 135-136). Are extracellular vesicles shed and taken up at the same time by VSMCs?
Lane 139: “calcium entry is in part via extracellular vesicles…” Do these vesicles fusion with the plasma membrane during “entry”? Is calcium accumulated in these vesicles and how? Please clarify.
Lanes 145-146 : “…VSMC can increase kinase activity in a frequency-dependent manner through trapping phenomenon.” Authors probably refer to the acquisition of autonomous kinase activity of certain CaM kinases. Please specify, and refer to the appropriate part of the Article.
Lane 191: “…Calmodulin…Calcium ions” : …calmodulin…calcium ions. And everywhere in the text: please use superscript for ion charges (Ca2+, Na+ etc.)
Fig.2.: Calmodulin structural changes upon calcium binding have been included. Please show mechanism of target enzyme activation (for example: the binding of the calcium-calmodulin complex to auto-inhibitory target protein regions leading to conformational change and target protein activation).
Lane 329: “Moreover its phosphorylation increase the force and kinetics of twitches…” : increases
Lanes 356-357: “The rise of intracellular Ca2+ in VSMCs promotes nuclear translocation of Ca2+-CaMKII. This event activates CaMKIV phosphorylation of CREB.” Please specify the mechanism, by which nuclear translocation of CaMKII activates phosphorylation of CREB by CaMKIV.
Lanes 478-479 : “Exerted cellular forces are transduced via the the cytoskeleton, and therefor of key importance in mechanosensing…” : ...and are therefore... And please delete second “the”.
Lanes 487-491: “Additionally, store-operated calcium entry (SOCE) is a central mechanism in cellular calcium signalling and in maintaining cellular calcium balance and linked to CaMKK2.” Please explain, by which mechanism is SOCE linked to CaMKK2.
Lanes 547-548: “whereas CaMKII inhibits CREB phosphorylating it.” : whereas CaMKII inhibits CREB by phosphorylating it. (Or: …whereas CREB is inhibited by phosphorylation by CaMKIII ?)
Lane 583 (Author contributions): “…and Kanin Wichapong…” : Please note that the names of the other Authors are abbreviated. Is Kanin Wichapong a co-author? If yes, why the name is not in the list of Authors (lanes 7-8)?
In conclusion, Authors are invited to further enhance the clarity of the Paper as detailed above, and are strongly encouraged to read their final text with the attention needed to correct eventual errors prior submission. Minor editing by a native English speaker would be also helpful for enhanced clarity.
